# Pain and Stress Detection Using Wearable Sensors and Devices—A Review

**DOI:** 10.3390/s21041030

**Published:** 2021-02-03

**Authors:** Jerry Chen, Maysam Abbod, Jiann-Shing Shieh

**Affiliations:** 1Department of Mechanical Engineering, Yan Ze University, Taoyuan 32003, Taiwan; s1088701@mail.yzu.edu.tw; 2Department of Electronic and Computer Engineering, Brunel University London, Uxbridge UB8 3PH, UK

**Keywords:** pain detection, stress detection, wearable sensor, physiological signals, behavioral signals

## Abstract

Pain is a subjective feeling; it is a sensation that every human being must have experienced all their life. Yet, its mechanism and the way to immune to it is still a question to be answered. This review presents the mechanism and correlation of pain and stress, their assessment and detection approach with medical devices and wearable sensors. Various physiological signals (i.e., heart activity, brain activity, muscle activity, electrodermal activity, respiratory, blood volume pulse, skin temperature) and behavioral signals are organized for wearables sensors detection. By reviewing the wearable sensors used in the healthcare domain, we hope to find a way for wearable healthcare-monitoring system to be applied on pain and stress detection. Since pain leads to multiple consequences or symptoms such as muscle tension and depression that are stress related, there is a chance to find a new approach for chronic pain detection using daily life sensors or devices. Then by integrating modern computing techniques, there is a chance to handle pain and stress management issue.

## 1. Introduction

Pain is a highly inter-variated and subjective feeling. What makes one person feel excessive pain may not be exactly same for another. In order to reach a general perception of pain, people have been constantly looking for a relevant scale or index try to quantify this sensation objectively for hundreds of years [1]. To extract more information that helps better understanding of pain, required numerous studies based on experiments and clinical observations. Since pain generated in both types of scenarios is linked to the same original sensation that is embedded in the human body, the mechanism of the pain is being clarified by conducting more and more experiments or observing the symptoms in the clinic. Back in 1846, when the first anesthetic (ether) was publicly demonstrated for general anesthesia by Morton at Massachusetts General Hospital in Boston (MA, USA), the majority thought the agony of pain has become history. However, looking back from now, that event might just be the beginning of our understanding of the mechanisms of pain. In general anesthesia during surgery, anesthesiologists use their knowledge of anesthetics to make subjects go into unconsciousness and block the sensation of pain for the purpose of performing the surgery more smoothly, but pain is a strong sensation that not only exists during surgery but also can exist potentially in any moment of our lives. It acts as more than just an unpleasant experience to everyone but also plays the role of a useful reminder to avoid potential injuries or tissue damage. Thus, pain research is not only about how to stop it, but more importantly what is the problem that this sensation is implicitly pointing to. Their causation of some pains is easy to identify, and thus can easily taken care of by treating the causative wounds or injuries. However, not all types of pain have a clear or obvious reason responsible for it. Other kinds of pain have no clear correlated injuries or wounds that need to be cured. Sometimes this type of pain is observed even after the original injuries have healed. Unfortunately, pain is also tricky to study for two reasons: pain-inducing tests are rarely viable in research; in fact, it is hard to design a pain-related experiment without ethical conflicts with human rights. For this reason existing pain-inducing tests (e.g., hot plate test, tail flick test, etc.) solely use animals as experimental subjects or focus on the relation between a pre-existing pain condition with some specific movement (e.g., back pain-inducing test [2] examines the pain condition during movements of lying supine, rolling over, and sitting up). The challenge of designing a proper pain-inducing test leads to the second difficulty, which is that pain detection is thus limited in the medical field or clinical aspects. Most pain-related research is done by observing patients in the clinics or during surgery. The devices used in measuring pain are usually expensive and only viable in hospitals and mostly used in surgery rooms. Thus, this review article aims first to emphasize the association between pain and stress and then, by adapting resourceful tests and wearable sensors-based detection techniques for stress detection, it aims to overcome the bottleneck of the universal pain detection problem. 

## 2. Review Scope

Different from previous reviews that were centered around the compliance and usability different among self-report pain scaling [3] such as Visual Analogue Scale (VAS), Verbal Rating Scale (VRS) and the Numerical Rating Scale (NRS) in clinical use, this review also brought up other pain scaling approaches that are based not solely on subjective self-reporting but rather objective physiological signals for pain detection. Such equipment is only applicable in hospitals for inter- or post-surgery studies. Its potential to be universalized by wearable sensors is discussed in this review. On the other hand, although reviews on the topic of stress detection [4] are often based on physiological signals and algorithmic approaches for feature extraction, wearable sensor systems for stress detection are rarely discussed with pain detection and its integration for health monitoring systems in mind. Furthermore, the difficulties of applying pain-inducing experiments urges this paper to dive into the relations between pain and stress, their mechanisms, correlations, assessment, and applied medical devices and wearable sensors. The purpose of this review is summing up the modern physiological and behavioral-based techniques for both pain and stress detection. Then we also discuss the demands for a wearable-based monitoring system, the evaluation of the system and its possibilities to overcome the issues of pain management and stress management.

## 3. Mechanism of Pain

The mechanisms of pain are being clarified by more and more studies and research. The pain process is coming to be understood as a dynamic phenomenon [5]. The nociceptive signal travels from receptors (nociceptors) to peripheral nerves then to the spinal cord and then to cerebral structures where the thalamus transmits the signals to the somatosensory cortex, frontal cortex and limbic system. Although the sensation of pain is being carried by nerve fibers [6], different types of nerve are used for different sensations as shown in Table 1. The first kind of nerve fiber is A-alpha nerve fibers; its diameter is about 13−20 μm long. Its signal conduction speed is about 80−120 m/s and it is in charge of carrying information related to position and spatial awareness. The second type of nerve fiber are called A-beta; its diameter is 6−13 μm and it conducts touching signals at a speed of 35−75 m/s. The third type of nerve fibers are A-delta ones; despite the fact they have a smaller diameter (around 1−5 μm) and slower speed for conducting signals (5−35 m/s), information such as sharp pain and temperature are delivered through them. The last type of nerve fiber is C fibers which have the smallest diameter at around 0.2−1.5 μm and the slowest signal conducting speed at 0.5−2.0 m/s but they can carry information such as dull pain, temperature, and itching. The different speeds of signal conduction may cause the sensation of a sequence of signals in subjects. For example, since t A-delta fibers are larger and surrounded by myelin (a lipid-rich substance that acts as an insulator for nerve cell axons), when someone pricks their finger, they are expected to sense the sharp sensation followed by a slower ache due to the difference of speed between A-delta fibers and C fibers.

On the path of transmitting pain to the brain, nerve fibers go through the dorsal horn that acts as a relay station or gateway for the signals [7]. Inside the spinal cord, the dorsal horn intervenes in the transmission of nerve signals; it either amplifies the nociceptive signal and pass it through or decreases the amplitude of the signal and ends it. This gate- like behavior, first proposed by Melzack and Wall, is known as the “gate control theory of pain” [8]. According to this theory, when a pain signal reaches the spinal cord and the central nervous system (CNS) it could be either amplified, reduced, or blocked by the system. This kind of condition is commonly observed in cases after subjects have experienced a severe injury and suffering paralysis of the lower limbs. The intervention of pain signals is also related to the types of nerve fiber. That is, signals carried by different nerve fibers have different priority in the sensation mechanisms at the spinal cord [9]. One instance is when people rub a wounded body part, which seems to attenuate the sensation of pain. This is due to the modulating effect of the counter-mechanism on large-diameter afferent fibers inhibiting the transmission and small-diameter afferent fibers facilitating the transmission [10]. Thus, when the A-beta fibers synapse is activated, it has the tendency to close the gate then mediate the sensation of C nerve fibers. However, there do exist some cases (e.g., phantom limb pain) that gate control theory alone cannot explain. It also involves the mechanism of the brain [8].

Finally, besides the interactions between nerve fibers and the spinal cord, there are other factors that may deviate the perception of pain. Factors that could affect the perception of pain are emotions [11] and psychological state [12]. Different mindsets and expectations toward the pain could either enhance the pain experience or reduce it. Personal beliefs and values under social or cultural influences may alter the perception of pain or vice versa [13]. Physical state changes (e.g., age, health status, etc.) could also worsen the perception of pain [14]. 

## 4. Classification of Pain

Generally, pain is evaluated in multiple aspects such as the location of the pain, the possible causes, the frequency of the pain occurrence, its intensity and the period of the pain. Classifying pain benefits the communication between patients and clinicians which hence facilitates the assessment task, helps formulate treatment planning and increases the precision of diagnoses in the clinic. In reality, however, not all pains have clear causes linked to them or have an adequate treatment for the pain. Some pains might have apparent causes but no adequate treatment (e.g., deep tissue disorders, peripheral nerve disorders, etc.) while pains like trigeminal neuralgia have adequate treatment without the causation being known. Then there are other pains that neither have clear causes nor treatment such as back pain and fibromyalgia [15]. Due to the complexity of pain causes and adequate treatments, there exists numerous methods for the classification of pain. Such classifications are expanding and new types of pain are being overserved in the clinic. The classification itself can sometimes ironically be confusing to clinicians [16]. However, there is a general consensus on pain classification that is agreed upon by a majority of researchers and clinicians.

### 4.1. Classification of Pain by Its Mechanisms

One of the most common classification techniques is based on the different mechanisms that originate the pain [17], which classifies it into the following types:

#### 4.1.1. Nociceptive Pain

Pain caused by injuries to body tissues is classified as nociceptive pain. This is the pain that comes after a cut, burn or fracture type of body tissue injury. Other pain of this type can commonly be observed in subjects who have undergone surgery during the postoperative period. This type of pain is described as aching, sharp or throbbing. Since the pain is caused by a body tissue injury, any movement (e.g., coughing, touching, etc.) related to the injured part can amplify the pain sensation. 

#### 4.1.2. Neuropathic Pain

Neuropathic pain originally meant the pain caused by a primary lesion, the dysfunction or transitory perturbation of nerves or the peripheral or central nervous system until it was redefined by International Association for the Study of Pain (IASP) taxonomy as “pain that caused by a lesion or disease of the somatosensory system”. Neuropathic pain is not a single disease; it is a syndrome caused by different diseases and lesions for which some of the underlying mechanisms might be unknown [18]. Sometimes neuropathic pain is depicted as a burning, tingling, and numbness sensation. People suffering neuropathic pain can also feel excessive pain from minor stimuli such as a light touch.

#### 4.1.3. Nociplastic Pain

Nociplastic pain are defined in the IASP 2017 taxonomy as “pain that arises from altered nociception despite no clear evidence of actual or threatened tissue damage causing the activation of peripheral nociceptors or evidence for a disease or lesion of the somatosensory system causing the pain”. Nociplastic pain is a relatively new term compared with nociceptive pain and neuropathic pain. In fact, there was only nociceptive pain and neuropathic pain before the IASP added the third mechanistic descriptor to its taxonomy. The call for the third mechanistic descriptor was to fill the lack of a proper valid pathophysiological descriptor among patient groups having fibromyalgia, complex regional pain syndrome (CRPS) type 1, or other instances of “musculoskeletal” pain and functional visceral pain disorders. As stated in [19]: “This group comprises people who have neither obvious activation of nociceptors nor neuropathy but in whom clinical and psychophysical findings suggest altered nociceptive function”. However, signs of this altered nociception have not yet been characterized by IASP [20] and it requires more studies on patients suffering from chronic pain. Furthermore, there is also a proposal for definition modification of nociplastic pain as “pain that arises from altered nociceptive function” in [21].

### 4.2. Classification of Pain by Its Time Period

The most common classification is concerned with the time duration of the pain. That is, by observing how long the symptom lasts, the pain could also be divided in two types: acute pain and chronic pain [22,23].

#### 4.2.1. Acute Pain

The term acute pain often refers to the occurrence of damage to tissues. It is a short-lived pain that works as a warning sign from the body. In most cases (i.e., broken bones, surgery, dental work, labor and childbirth, cuts, burns), the pain last fewer than six months and disappears once the injury or disease is cured or healed. 

#### 4.2.2. Chronic Pain

The features of chronic pain (e.g., osteoarthritis, frequent headaches, low back pain, etc.) are its long duration and its complicated mechanism. It usually lasts for more than six months, even after the original injury was healed. Due to its stubbornness, people living with chronic pain may develop symptoms of anxiety, depression, or other conditions (e.g., tense muscles, lack of energy, limited mobility, etc.).

## 5. What Is Stress and what is Its Correlation with Pain?

Pain and stress are interleaved and connected in many ways and the consensus of the highly intertwined relations between these two mechanisms has been established in [24]. Stress is a feeling of emotional strain and psychological stress and a state of threatened homeostasis and a reaction that breaks the balance of physiological processes. The sources of stress could be the experience of the pain or the consequences of ongoing pain or other psychological reasons. Experiencing pain is stressful enough, especially when it is not negligible, but what is worse is when the pain lasts for a longer time, then it could lead to a vicious cycle. For example, people suffering back pain could easily develop stress by further induced muscle tension or spasms [25]. The muscle tension produces more pressure on the nerves, not only causing more pain and stress but also squashing the nerve harder and exacerbating the pain. On the other hand, the consequences of ongoing pain usually last longer than half a year and this kind of chronic pain would have a greater impact to the patient’s quality of life [26]. The stress and frustrated mood brought by the chronic pain are already tough, but the restricted movement or low physical activity in fear of amplifying the pain furthermore are even worse in these situations. People in the fear of being in pain tend to avoid any potential movement that does or may induce the pain. The avoidance and anticipation of pain that causes a lot of stress is the beginning of the vicious cycle. This kind of symptom are called “pain catastrophizing” [27]. It is a negative cognitive-affective response toward actual pain or the anticipation of pain. Furthermore, experiencing stress could also affect the endocrine system balance and then induce endocrine disorders which are linked back to chronic pain [28].

Upon encountering stress, the human body would respond with three components: adrenal medulla, hypothalamus and pituitary gland. These three components constitute the so called hypothalamic-pituitary-adrenal axis (HPA) which react to the stress by releasing hormones (the adrenal medulla could release norepinephrine) helping or exciting other parts of the organism through the sympathetic nervous system (SNS) [29]. When the SNS is activated, the subject’s heart rate and blood pressure would increase in a short period of time; their breathing may get faster, adrenalin levels raise as do the blood sugar and cholesterol levels. The blood flow would also be redirected from lower priority organs such as the organs in the digestive system to higher priority vital organs such as the heart and the brain. The function of the immune system would be driven up since there is an immediate danger and the body needs to handle the “fight or flight” situation [30]. However, if this situation cannot be resolved immediately (e.g., chronic pain), this self-protecting mechanism might harm the body instead and becomes maladaptive in the long term. Excessive or prolonged activation of the SNS causes muscle tension, headaches, high blood pressure or even promotes the development of cancer [31]. People who are physically inactive due to a stress state could rather end up with depression.

## 6. Assessment for Pain and Stress

### 6.1. Pain Assessment

Pain is gradually being accepted as the fifth vital sign [32] since this was firstly proposed by American Pain Society (APS) in 1996. Different kinds of pain (i.e., acute pain or chronic pain) are assessed separately and serve different purposes. The assessment for acute pain is to avoid provoking the pain onset and to monitor the effect of the suppressant that is used. Contrarily, the goal of assessing chronic pain is collecting related signs in the early stages or to gather enough symptoms to track down the origin of the pain. In practice, there are multiple scales and measures that are helpful for tracking pain-related treatment outcomes. These kinds of measurement are resources for clinicians to select a treatment plan and validate the treatment effects. Commonly used measures for pain are: (1) Self-report measures: self-report measurement is a subjective score related to pain given by the subject ranging from 0 (no feeling of pain) to 10 (extreme pain). It usually refers to a numerical pain rating scale, and similar measurements are VAS [33]; (2) Physical performance tests: the 5-minutes walking, stair-climbing task, 15 meters walking, sit-to-stand and loaded forward-reach test [34] and the Abbey Pain Scale for the non-verbal individuals (e.g., patients with dementia) [35]; (3) Physiological response measures: the physiological and autonomic response measures are the most objective and physiological approach to pain. By observing the changes of multiple physiologic signals such as skin conductance and heart rate and other signals, researchers can formulate a valid index for pain evaluation (e.g., analgesia nociception index [36]). However, the correlation of such measurement with pain are still under debate. Since the idea of using such measurement is to apply the physiological signals that are the subject of the activation of the automatic nerve system. However, the activity of the automatic nerve system (mainly about the balance of sympathetic nervous system and parasympathetic nervous system) may be reacting not only to pain but also other factors. Thus, the physiological measurement in most cases is used in the surgical room where the subject is unconscious so the physiological approach is the only way to obtain any relevant information for pain monitoring.

To date, numeric pain scales based on the patient’s self-reporting is still the easiest and most popular assessment for pain. However, the lack of an objective assessment for pain may cause the overuse of opioids and to their addiction in clinic. This problem could further lead to opioid-related unintended deaths [37]. 

### 6.2. Stress Assessment

Similar to pain assessment, one way to assess stress is by a self-report scale in a clinical environment. Rather than a subject filling out a questionnaire, the VAS provides a rapid quantitative assessment in a 10-points range [38]. Since stress is defined as a state in which homeostasis is threatened, the adaptive processes that are activated would cause both physiological and behavioral changes. In order to comprehend this stress response mechanism, numerous studies have been conducted observing the physiological and behavioral changes in the body under stress induction tests. The observation of physiological or pathophysiological changes in response to stress is fundamental to the development of novel pharmacological agents for stress management [39]. In the rapid development of modern society, people are dealing with stress and work fatigue on a daily basis; thus, like pain assessment, stress assessment could also benefit from observation during daily life. Without further inducing stress to the subjects, such cases are associated to fatigue and work-related stress [40] in the concern about the mental and physical health of employees.

#### Stress Induction Tests

Setting up stress-inducing scenarios helps researchers collect and validate the stress- correlated physiological signals or behavioral signals. These stress induction tests usually involve asking the subject to finish a certain task or perform a certain action in a specific condition designed by researchers; then the researchers could conclude which signals are related to stress by monitoring the changes of signals during the tests.

Trier Social Stress Test

The original Trier Social Stress (TSST) consists of an anticipation period and a test period for 10 minutes each [41]. During the test, subject is told to take role of a job applicant and prepare for a 5-minutes speech. An audience of three persons plays as interviewers and managers. The subject must convince the interviewers of his/her suitability for the imaginary job without touching any topics that is previously noted before the test. If the subject finishes his/her presentation early, he/she will be asked to continue by the interviewers. Then after the speech period is over, the subject is asked to do a mental arithmetic which is counting down numbers from 1,022 in steps of 13. Once the subject makes a mistake, he/she will need to start all over from number 1,022 again. Following the mental arithmetic test, the subject is given details of the experiment and will be allowed to take a rest while his/her physiological signals are still under monitoring. There are other kinds of TSST variations such as using the virtual reality (VR) to reduced cost [42].

Stroop Color-Word Inference Test

The idea of the Stroop color-word inference is asking the subject to read out the color of the words while that word is printed in different color of the word is represented literally. The stress is induced by the contradiction between the linguistic and visual perceptions. The Stroop color-word test has been widely used in psychology for a long time [43]. It has the advantages of high reliability and stability in measuring for individual differences with only relatively simple rules. The performance of Stroop color-word test also has a positive outcome in VR environment [44].

Cold Pressor Test/Hot Water Immersion Test

In the cold pressor test, the subjects are asked to put their hand into a bucket of cold water and keep there it as long as they can. The subjects should notify the researcher when they first feel that the cold water starts causing pain to their hands. Then at any time after the first notification is given, the subjects are free to remove their hands when they feel the pain is unendurable. Then according to the timing of two notifications (i.e., when does the subject start feeling pain and when they remove their hands due to the intolerable pain) and the continually collected blood pressure and heart rate, researchers can further analyze the physiological features of stress. The cold pressor/hot water immersion test are basically the same and the only different is the temperature of the water that is being used in the test. The cold pressor test is efficient experimental stress induction [45] which has been observed to reliably increase HPA activity [46].

International Affective Picture System Test

In psychological studies, one of the most common tests for emotion and attention research is the International Affective Picture System (IAPS). By providing pictures ranging from simple daily objects to extreme pictures that involve violent or erotic contents, the test induces stress or emotion in the subject. A relevant application is used to detect IAPS stress levels in human pilots [47].

### 6.3. Physiological Signals for Assessment

#### 6.3.1. Heart Activity

Since stress causes fundamental disturbances in the autonomic nervous system (ANS) which has major effects on heart activity [48], some useful detection methods for stress are based on heart-related signals [49]. Heart activity could be represented by an electrocardiogram (ECG), which is recorded by measuring the electrical activity of heartbeats. Usually, a normal heartbeat includes three distinguishable waves: the P wave, QRS complex wave and T wave. Most of the studies on heart activity are related to three aspects of the heart: time domain, frequency domain and non-linear features of heart. The research in the time domain focuses on parameters such as heart rate (HR), inter-beat (RR) intervals and heart rate variability (HRV) [50]. For RR intervals, it could be further studied as its mean value, the standard deviation or root mean square. Frequency domain studies analyze the components in the low-frequency (LF), high-frequency (HF) or the LF/HF ratio. As for the non-linear features there are algorithms such as entropy, complexity, Poincare Plots [51], recurrence and fluctuation slopes.

#### 6.3.2. Brain Activity

The brain activity is recorded as the electroencephalogram (EEG) for brain-related research (e.g., emotion changes, stress-related studies [52] or consciousness studies for anesthesia, etc.). The four bands of the EEG signal are alpha (8−13 Hz) which indicates the sign of calmness and balanced state of mind; beta (13−30 Hz), which is related to emotional and cognitive processes which correlates to stress; delta (0.1−4 Hz) which is associated with deep sleep stages (e.g., high brain activity in this range are being viewed as a sign of unconsciousness) and theta (4−8 Hz) that generates the theta rhythm which is a neural oscillation in the brain that is linked to interpretation of cognition [53] and behavior such as learning, memory and spatial navigation.

#### 6.3.3. Muscle Activity

Muscle tension usually comes along with stress. Researchers are studying changes of muscles activities in human body under certain stress-presented activities [54]. The state of muscles like stretching and releasing could be monitored by electromyogram measurements. Electrodes placed on certain areas of muscle could detect the potential changes due to the locomotion of the body. After obtaining the measurement of muscle activity, statistical techniques can be used to enhance the understanding of the signals. Such applications are often referred to “myomonitoring” and can be adopted by studies in the interest of muscle tension (e.g., monitoring mandibular closure maximum intercuspation of the teeth [55]), and muscle fatigue (with a sonic approach [56]).

#### 6.3.4. Electrodermal Activity

EDA is a useful indicator for neurocognitive stress by giving the change of electrical properties of skins. When a stress-inducing scenario is applied on a subject, the body is expected to start sweating; this further increases the skin conductance [57]. The long-term shifts in tonic level are called skin conductance level (scl); and the transient responses within seconds are the galvanic skin response (GSR). The tonic and phasic measurement are the two main aspects of EDA. One example of using EDA to validate pain stimulation is given in [58]. Posada-Quintero et al. proved that thermal grill stimulation is highly correlated with VAS. In this research, the observed EDA also shows significant increases as the stimulation level goes up. A systematic review of EDA data collection and signal processing presented in [59] by Posada-Quintero and Chon provides a summary of EDA recording devices, signal analysis methods, and the synthesis framework for EDA-related research.

#### 6.3.5. Blood Volume Pulse

Blood volume pulse (BVP) provides the changes of volume in blood between each heartbeat and it fluctuates along with the changes in heartbeats. BVP is measured by optical, non-invasive sensors by comparing the light absorbed by the blood. Xie et al. used BVP for identifying strong stress and weak stress [60]. 

#### 6.3.6. Skin Temperature

Stress influences both the core and peripheral body temperature [61]. While the core temperature tends to rise in response to stress, the distal skin locations tends to decrease. When acute stress is present, it triggers peripheral vasoconstriction which causes a rapid drop in skin temperature [62]. 

### 6.4. Behavioral Signals for Assessment

#### 6.4.1. Speech

Voice patterns can be quite different for a person under stress or not. Multiple features related to the voice patterns could be altered when stress is present such as the changes in pitch [63], tone and speaking rate, or even the words they choose in the speech.

#### 6.4.2. Facial Expressions

Natural habits, such as facial expressions are the reflection of the psychological state and the indication of emotion that a person is experiencing. Lots of researchers are trying to capture these subtle-signs and correlate them with stress situations through facial electromyography (EMG) [64] or image recognition based on facial expression [65].

#### 6.4.3. Keystroke and Mouse Dynamics

In modern times, most people use computers on a daily basis. The way they use them could provide clues about their mental state or emotions in the present. The speed of typing and mouse dynamics could provide useful clues to indicate whether a person is stressed or not [66,67]. The use of excessive strength when hitting a keyboard and clicking a mouse is no doubt an obvious sign of an upset mind.

#### 6.4.4. Body Gestures and Movements

A stressful person also displays signs of their state of mind with their body actions such as jaw clenching, constant finger rubbing, or even posture changes while they are standing or sitting. Multiple behavioral features are provided in [68] for stress detection.

#### 6.4.5. Mobile Phone Usage

A person experiencing stress could either choose to fight it or flee from it. Using a cell phone might provide an easy way to forget about the stress. By distracting themselves with various features built into a smartphone, the stress may seem to fade away temporarily. Mobile phone addiction could also be a sign of anxiety [69] which is common symptom for people under stress. One study [70] finds a significant correlation between mobile phone use and stress.

#### 6.4.6. Questionnaires and Surveys

Questionnaires and surveys are already being widely used in psychological research for assessment of psychological state. By asking subjects questions that serve to identify a specific mental approach, subjects might expose their deepest concerns or stress on their minds. Sometimes, even the subject could not know the source of their own stress which requires questionnaires or consultation by a psychologist to unravel.

## 7. Medical Devices or Wearable Sensors used in Pain and Stress Detection

In this section, the common devices and wearable sensors that are suitable for detecting pain and stress are organized. Pain within a short time-period usually can be located by the person themselves or be detected by a clinician in a clinical environment. Even in surgery when the subject is unconscious, there are devices to provide a valid index for the degree of pain being experienced [71]. However, sometimes pain like chronic pain or stress are intermittent rather than constant. This features highly irregular seizure timings which are hard to detect in a specific time window using traditional medical devices or clinical assessments; in fact, relevant diagnoses are mostly dependent on self-reporting which is based on the subject’s memory to get any relevant information. In this case, wearable sensors could be used to collect data when subjects are not in the clinic and monitor both physiological and behavioral signals for longer periods which provides useful information for clinicians [72]. 

### 7.1. Medical Devices Used in Pain Detection

Nociception is the most relevant and effective approach for pain detection. Even though pain is a subjective perception, nociception is a physiological reaction to the nociceptive stimuli; this nociceptive stimulus is based on the reaction of the autonomic nervous system (i.e., the balance of the sympathetic nervous system and parasympathetic nervous system). Then by analyzing the corresponding physiological signal variation caused by the activity of autonomic nervous system, researchers manage to formulate an index useful as a pain reference. A popular way to monitor the balance of the sympathetic nervous system and parasympathetic nervous system is by analyzing the heart rate variability. Other methods use the number of skin conductance fluctuations per second (NFSC), the size of pupil and its variability when illuminated, blood vessel contraction, EMG, EEG, and changes of body temperatures, etc. Heart rate variability and plethysmography are widely used in both types of research since these two signals are easier to obtain during surgery and they are highly sensitive to the activity of autonomic nervous system. The two most common medical devices used in assessing pain are the analgesia nociception index that is based on the heart rate variability and the surgical pleth index that is based on plethysmography.

#### 7.1.1. Analgesia Nociception Index

The analgesia nociception index (ANI) is a technology that provides a measurement of the parasympathetic tone on a scale from 0 to 100. It has been used in the surgical room or post-operative room for pain assessment. By analyzing the electrocardiographic data which reflecting parasympathetic activity, the ANI provides a reference to the sympathetic/parasympathetic balance that allow doctors to control surgical stress. The ANI has shown a correlation to the self-rating system in the postoperative period after volatile agent and opioid-based anesthesia in [73]. Since this index is solely based on the physiological signals rather than being a self-rating system, it could be applied to patients under general anesthesia or in critically ill condition who have communication problems. Besides, in a study of ANI in pain-related surgical conditions (e.g., during labor [74], laparoscopic abdominal surgery [71]), there is also other research such as [75] that find the relations between ANI and emotional status. Basically, mechanisms that are correlated to parasympathetic changes could adopt ANI as a measurement tool.

#### 7.1.2. Surgical Pleth Index

The surgical pleth index (SPI) is a digital monitor based on the patient’s hemodynamic responses to surgical stimuli and analgesic medications during general anesthesia [76]. SPI measures the sympathetic activity as a reaction to painful stimuli; it creates a single index (using a scale from 0 to 100) by integrating the photoplethysmographic amplitude and photoplethysmographic pulse interval with algorithms. The SPI has been proved to correlate with self-reporting systems [77]; however, the appropriate selection of SPI target values has not been established yet. The representation of different score ranges of SPI (e.g., prediction of moderate-to-severe postoperative pain [78]) is still a major topic in the research field.

### 7.2. Wearable Sensors Used in Stress Detection

Assessments for stress detection could benefit from the proper use of wearable sensors for data collection based on the physiological/behavioral signal of interest. The useful physiological signals that are of interest in stress detection research are heart activity (ECG), brain activity (EEG), muscle activity (EMG), skin conductance (EDA), BVP, and skin/body temperatures and relevant wearable sensors and devices for stress detection are organized in Table 2. Most of the behavioral signals could be obtained by smartphone sensors [79] or recorded by video cameras for image analysis and voice analysis. Each sensor used in stress detection is listed in the following paragraphs, with additional details of the sensor placement illustrated in Figure 1.

Empatica E4 wrist band: this device is a wrist band is a real-time physiological data streaming and visualization sensor. As a medical-grade wearable device, it enables researchers to collect multiple physiological data such as BVP for HRV analysis, and EDA that reflects the constantly fluctuating electrical properties of a certain area of skin and peripheral skin temperature. Besides, it also captures motion activity with a 3-axis accelerometer [80,81,82,83].AutoSense: this is a wireless sensor suite that packs six sensors in a small form factor which are capable of collecting cardiovascular, respiratory and thermoregularity measurements through radio transmission and processes collected signals for detecting the general stress state of subjects. The wearable sensor has advantages of excessive lifetime while fully charged which allows prolonging its use for constant data collection [84,85,86].SleepSense: this is a belt-like sensor which adopts a piezoelectric film sensor for converting chest or abdominal respiration motions to analog voltages and thus, provides an indication of respiration waveforms [87,88].BN-PPGED: this is a physiological sensor for measuring BVP via optical plethysmographic methods and EDA activity. The sensor could be worn as a wristband with an additional two electrodes situated on two fingers [89].Cardiosport TP3: this is also a belt-like wearable sensor. By attaching the sensor pod to the chest strap, the TP3 will be activated to collect HR and millisecond RR intervals as long as the HR is detected [90].Q-sensor: this is a wireless sensor designed by the Massachusetts Institute of Technology that aimed to “detect and record physiological signs of stress and excitement by measuring slight electrical changes in the skin.” The emotion detection sensor could benefit individuals with autism who usually do not show his/her stress outward and helping to manifest the emotions before breakdown. The sensor could obtain the accelerometer data and skin conductance by measuring inner wrists of subject’s hand [70].Wahoo chest belt: Wahoo chest belt is equipped with a sensor which collects HRV data on a chest belt. Besides provides the heart rate and calorie burn data for workout evaluation, the HRV data could also be an indicator of the autonomic nervous system activity [91].BioHarness 3: this is physiological monitoring telemetry device that are usable for subjects in the workplace. The device can store and transmit data such as HR, HRV, respiration rate, and 3-axis accelerometer data through Bluetooth [92].Shimmer sensor: the shimmer sensor is a monitoring wearable sensor for EDA. Composed of two finger electrodes and a main unit, the shimmer sensor can transmit data to personal computer or other devices through Bluetooth connections [92].MindWave mobile EEG headset: it is an EEG headset capable of logging single channel EEG raw data at a 512 Hz sampling rate then provides index of attention and meditation of the user after power spectral density analysis [92].DataLOG: this is a portable EMG signal collection and monitoring devices designed by Biometrics. It could be placed on the arm, the leg or waist for various fields studies like human performance, sports science, medical research, industrial ergonomics, gait laboratories, and educational settings [93].

**Table 2 sensors-21-01030-t002:** Wearable sensors used in stress detection.

Type of Signal	Commercialized Wearable Sensors Used in Relevant Research	Wearable Sensors Not Yet Commercialized but Used in Relevant Research
Heart activity	Empatica E4 wrist band, AutoSense, Cardiosport TP3, Wahoo chest belt, BioHarness 3	
Brain activity	MindWave mobile EEG headset	Device 1, Device 2
Muscle activity	DataLOG	Device 3
Electrodermal activity	Empatica E4 wrist band, BN-PPGED, Q-sensor, Shimmer sensor	
Respiratory	AutoSense, SleepSense	
Blood volume pulse/pulse plethysmograph	Empatica E4 wrist band, BN-PPGED	
Body/skin temperature	Empatica E4 wrist band, AutoSense	
Three-axis accelerometer data	Empatica E4 wrist band, Q-sensor	

Notes: Empatica E4 wrist band is used in [80,81,82,83]; AutoSense is used in [84,85,86]; SleepSense is used in [87,88]; BN-PPGED is used in [89]; Cardiosport TP3 is used in [90]; Q-sensor is used in [70]; Wahoo chest belt is used in [91]; BioHarness 3, Shimmer sensor, and MindWave mobile EEG headset are being used as an integrated system for stress monitoring in [92]; DataLOG is used in [93]; Device 1 is a EEG wearable sensor developed in Online Predictive Tools for Intervention in Mental Illness (PTIMI) project funded by European Union [94]; Device 2 is a noninvasive physiological sensor for stress assessment presented in [95]; Device 3 is used in [96] which they collect the EMG signals of the left trapezius muscle and then remove the contained ECG signal components.

**Figure 1 sensors-21-01030-f001:**
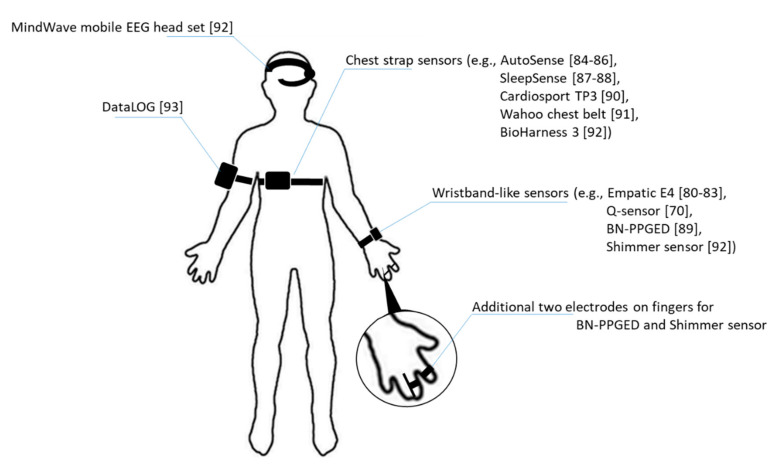
Illustration of sensors placement on human body.

## 8. Wearable Sensors in Healthcare

Wearable sensors are under rapid development and have been applied in many fields through the years, especially in healthcare. Recently, due to the severe pandemic, wearable sensors are also being used for COVID-19 detection [97]. The expanding needs of wearable sensors in healthcare domain is motivated by the increasing healthcare costs; the success application of wearable sensors in such domain is the result of advanced technologies of microelectronics and wireless communication [98]. The increasing healthcare costs are highly related to the ageing world population [99]. Constant monitoring for physiological and psychological advantages chronic diseases prognosis and detection in the early stage. A real-time feedback information about subject’s health status could greatly improve the accuracy and capability to identify abnormal condition and prevent it in advance. Wearable sensors are the applicable solution for such tasks; in fact, a type of system based on wearable sensors are called the wearable health–monitoring systems (WHMS) which can be adopted for supervising subjects that are elderly people, have chronic disease or special abilities [100].

According to Alexandros et al. [98], WHMS are founded with a base of various types of miniature sensors, wearable or implantable for the purpose of measuring physiological signals. The collected parameters are transmitted through wireless or wired link to a central node (e.g., microcontroller board) to processing and display the information to the users. Then, these aggregated vital signs can be sent to the medical center for further analyses and diagnoses by medical professionals. A WHMS has multiple components (i.e., sensors, wearable materials, smart textiles, power supplies, communication modules, central processing unit (CPU), software and advanced algorithms, etc.). Yet, it also has to meet several criteria for practical use such as light weight and small size in order to enhance its comfortableness, and low radiation and mild heat dissipating to ensure safety. Finally, appropriate security during the data transmission for privacy concerns. After all of the above requirements are satisfied, the overall WHMS also has to be low-cost and affordable for majorities or even the underprivileged minority. Several available WHMS were reviewed by Alexandros and Nikolaos [98] according to the maturity level of the system. The maturity level of system is evaluated to their ability to measure multiple parameters, the detail level of the data documentation, the popularity of the system based on its citation number, the applied hardware technologies in the system and the algorithms used for feature extraction and decision support. A maximum maturity level WHMS also must face problems like battery lifetime shortage and private information security. However, integrating with modern techniques such as cloud computing, fog computing resources, the WHMS can further improve their performance with minimum cost on electronic components, processing unit and even save more space for individual sensors. Such integrations [101] release the frontier data collecting unit from massive computing tasks which may free more space and drop power consumption to the sensors. Then using smart phone as data transmission and processing relay station to construct a more comprehensive and compatible platform, and with federated learning an explicit data could be reproduced to implicit data which reduce the risk of data leakage and accelerate the process for optimizing diagnosis model. 

## 9. Discussion

Pain and stress are useful mechanisms that help humans survive and is also a part of evolution. With the proper induction of the unpleasant feeling, individuals could sense the danger that is happening or might be harmful to its body; it is especially useful when the danger that caused the threat is beyond the individual’s knowledge. Pain and stress serve as warning signals to acknowledge any incidents that are potentially harmful or fatal. This mechanism also prevents the same or similar incidents that might happen again by introducing stress before the harmful incident can cause further damages. This kind of experience based on threat learning could help to address threats and are useful but not harmful to the body in the short term. However, under the rapid development of human society, these mechanisms for survival instinct are no longer needed as much as they used to be; contrarily, the downside of these two mechanisms in the long term has gradually surfaced. Pain-inducing tests are much less effective compared to stress-inducing tests, and any experiment that may induce pain affects human rights. Thus, a possible solution for pain monitoring might rely on more research to find out more details about the relationship between pain and stress, then using WHMS is expected to resolve the pain management issue.

Pain management is a raising issue worldwide [102]. The access to pain management has been defined as a human right, despite the differences in social status and economic condition. Everyone should be able to be free from suffering pain, but in reality, it is truly sad that not everyone could afford a physical examination in the hospital or treatment resources. Pain management is also a public health issue [103]. In addition, in developed countries, the aging population also must face chronic age-related diseases [104]. On the other hand, stress management also demands people’s attention as it is a common issue in modern society. Stress management techniques and relevant education are all necessary for students [105] and for workers [106], for everyone to meet their needs.

Fortunately, with the help of well-developed techniques, devices for monitoring pain or stress are becoming more and more accessible. The use of wearable sensors may allow the diagnose of pain and stress no longer restricted to hospital but everywhere by online doctors or artificial intelligence (AI) models.

## 10. Conclusions

Pain is an annoying feeling that everyone must have experienced; yet it is a subjective sensation to everyone. What is considered painful by one person might not be interpreted the same by others. The mechanisms of pain are so far being understood as a signal that travels from receptors to peripheral nerves, to the spinal cord, then to cerebral structures. Different nerve fibers have their own priority and duty for carrying sensations. There are a few classifications for pain according to their characteristics and mechanisms. One way to classify pain is by using the time-period of the pain duration. If the pain lasts less than 6 months, it is called acute pain; otherwise, if it lasts more than 6 months, it is known as chronic pain. Acute pain usually comes with a specific causation and the perception of it is constant until the causation to the pain has been removed or the causing injury healed. Chronic pain, on the contrary, does not affects the subject for a much longer time but the symptoms are often intermittent, which raises challenges to detect or to find the origins of that pain. Moreover, having a persistent pain issue leads to the development of stress which is also hard to detect or treat. A person suffer either from pain or stress could end up with both, due to the vicious cycle. Luckily, with multiple physiological signals and behavioral signals collecting by wearable sensors, there is hope for detection to seek moderate treatment in the early stage. Available physiological signals for stress detection on wearable sensors are heart activity, brain activity, muscle activity, electrodermal activity, respiratory, blood volume pulse, skin temperature. Furthermore, wearable sensors canwork with multiple components (e.g., communication modules, CPU, advanced algorithms, etc.) to construct a wearable health-monitoring system for chronic disease detection and health status monitoring.

This article has presented the mechanisms of pain and stress, the correlation between them, their assessment, and their detection devices as well. Finally, wearable sensor-based health-monitoring systems are presented and discussed in the hope of solving the imbalanced resources global-wide for diagnosing pain and pain treatment issues. The low cost and easy to use features of wearable sensors might provide a perfect solution for this. Awareness about the importance of pain management is rising along with the promotion among humanity. Integrated with AI algorithms and cloud computing resources, wearable sensors could act as more than a component that collects data but as a foundation of a health monitoring and treatment system. Furthermore, by analyzing and quantifying pain and stress, they provide an opportunity to deal with the worldwide issues of pain t and stress management. 

## Figures and Tables

**Table 1 sensors-21-01030-t001:** Different kinds of nerve fiber.

	A-alpha	A-beta	A-delta	C
Myelinated/unmyelinated	Myelinated	Myelinated	Myelinated	Unmyelinated
Size (diameter)	13–20 μm	6–13 μm	1–5 μm	0.2–1.5 μm
Speed of signal transmission in meter per second	80–120 m/s	35–75 m/s	5–35 m/s	0.5–2.0 m/s
Related perception	Position and spatial awareness	touching	Sharp pain and temperatures sensation	Dull pain temperatures and itches

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
