# Peer review of "Pain and Stress Detection Using Wearable Sensors and Devices—A Review"

_sensors, 2021, doi:10.3390/s21041030_

Round 1

Reviewer 1 Report

In this review, the authors give a review about the wearable sensors and devices to estimate the pain and stress. However, we think some parts need to be discussed more before the publication.

  1. The review should focus more attention on the device. Authors have discussed a lot about the pain and stress, it looks more like a medical review.
  2. In the “6.2 Wearable sensors used in stress detection”, we think the author should list the reference or production website after each introduction.
  3. To make the review more visual, we think some figures about the devices are helpful.

Author Response

Journal: Sensors (ISSN 1424-8220)

Manuscript ID: sensors-1070974

Type: Review

Title: Pain & Stress detection using Wearable Sensors and devices – A Review

Authors: Jerry Chen, Maysam Abbod, Jiann-Shing Shieh

Comments of reviewer I:

In this review, the authors give a review about the wearable sensors and devices to estimate the pain and stress. However, we think some parts need to be discussed more before the publication.

  1. The review should focus more attention on the device. Authors have discussed a lot about the pain and stress, it looks more like a medical review.
  2. In the “6.2 Wearable sensors used in stress detection”, we think the author should list the reference or production website after each introduction.
  3. To make the review more visual, we think some figures about the devices are helpful.

Authors responds:

  1. The review should focus more attention on the device. Authors have discussed a lot about the pain and stress, it looks more like a medical review.

Ans:

Thank you for your comment. We have added a new section to presents the wearable sensors in healthcare, the wearable health-monitoring systems (WHMS) and its integration to modern techniques in lines 526-568.

  1. In the “6.2 Wearable sensors used in stress detection”, we think the author should list the reference or production website after each introduction.

Ans:

Thank you for your suggestion. We have added references to each wearable sensors in lines 479, 484, 487, 490, 493, 499, 502, 505, 508, 511, and 515.

  1. To make the review more visual, we think some figures about the devices are helpful.

Ans:

Thank you for your comment. We have added a new figure for sensors to illustrate the sensor placement on human body in line 525.

Reviewer 2 Report

This contribution reviews research on pain/stress management, with regard to body pain mechanisms and its’ stress relations, furthermore to diagnostic procedures, instruments and wearable devices and sensors for data collection.

Paper is well organized and presented. However, a few patents, hardware solutions and psychological tests, previously published and presented in academic and commercial writings, are not included (fatigue related stress, e.g. in papers considering biomechanics in general; furthermore e.g. MyoMonitor etc). It shouldn’t be considered as an obstacle or a major flaw for accepting this paper. It just could have improved the scientific soundness and impact of authors’ effort.

There are many minor language and style mistakes that have to be corrected.

Author Response

Journal: Sensors (ISSN 1424-8220)

Manuscript ID: sensors-1070974

Type: Review

Title: Pain & Stress detection using Wearable Sensors and devices – A Review

Authors: Jerry Chen, Maysam Abbod, Jiann-Shing Shieh

Comments of reviewer II:

This contribution reviews research on pain/stress management, with regard to body pain mechanisms and its’ stress relations, furthermore to diagnostic procedures, instruments and wearable devices and sensors for data collection.

Paper is well organized and presented. However, a few patents, hardware solutions and psychological tests, previously published and presented in academic and commercial writings, are not included (fatigue related stress, e.g. in papers considering biomechanics in general; furthermore e.g. MyoMonitor etc). It shouldn’t be considered as an obstacle or a major flaw for accepting this paper. It just could have improved the scientific soundness and impact of authors’ effort.

There are many minor language and style mistakes that have to be corrected.

Authors responds:

  1. Paper is well organized and presented. However, a few patents, hardware solutions and psychological tests, previously published and presented in academic and commercial writings, are not included (fatigue related stress, e.g. in papers considering biomechanics in general; furthermore e.g. MyoMonitor etc).

Ans:

Thank you for your comment. The fatigue related stress has been added in lines 271-275; myomonitor is also added in lines 355-358.

  1. There are many minor language and style mistakes that have to be corrected.

Ans:

Thank you for your comment. The entire article is review again by all of the authors to rectify the linguistic errors. Particularly, one of our authors (i.e., Dr. Maysam F. Abbod) who is a senior reader in Brunel University London and has been living in UK for over 25 years. He has checked this revised manuscript not only the structure and contents but also line by line for English grammar and spelling.

Reviewer 3 Report

The authors have presented their review about Pain & Stress detection using Wearable Sensors and devices

The authors have put forward a fine effort in presenting their review. The article highlights pain detection and management, and applications of wearables for the same.

It is generally difficult to find novelty and innovations in a review article, and this article is not an exception. The review methodology is unclear and the objectives, goals, and purpose of this article are not explicitly understandable as the title and abstract seem to contradict. For example, the title suggests that this article is about a review of wearable sensors for pain / stress detections, whereas, the article is mostly about pain management with very little emphasis on the wearable sensors or at-least the non-invasive methods.

Below are my comments to the authors; and in the view of maintaining the scope and quality of the Journal, there are a few major concerns that must be addressed before this article can be accepted for publication.

Below are the comments and suggestions to improve the article:

  1. The presence of numerous other articles (original research / methodological reviews / systematic reviews / other reviews) limits this article’s relevance and novelty. More precisely, this article can be more comprehensive and should be expanded further to include relevant information and cover all the bases that touch upon the wearables. For instance, the wearables part consists of around 30% percent of the whole article. This should be expanded substantially since this is a review article. The scope and extent of the comprehensiveness and coverage could be improved to make this article stand out from the other similar reviews. This will provide the readers with a more in-depth understanding and clarity.
  2. As indicated in the multiple references within the article and numerous other articles published already, there are numerous other approaches to review the similar scope of work. Please include a section to emphasize how the approach of this review is different from others.
  3. Please make sure all the figures have references/citations. (throughout the article)
  4. Please make sure references/sources are adequate and correctly used throughout the article. (throughout the article)
  5. Please review usage of abbreviations/acronyms, and provide an explanation where required. (throughout the article)
  6. The article is missing a section about review methodology. A review methodology section is required to show how the literature review was conducted with an emphasis on eligibility criteria of the articles reviewed vs. the ones included in this review, search strategy, screening procedure, quality assessment, etc.
  7. There are several grammatical errors, dangling modifiers, sentence structure errors, etc throughout the article. The whole article should be thoroughly revised and proofread to rectify these errors. (throughout the article)
  8. Please rewrite sections 7 and 8 (discussion and conclusion) to reflect the above-mentioned corrections.

Author Response

Journal: Sensors (ISSN 1424-8220)

Manuscript ID: sensors-1070974

Type: Review

Title: Pain & Stress detection using Wearable Sensors and devices – A Review

Authors: Jerry Chen, Maysam Abbod, Jiann-Shing Shieh

Comments of reviewer III:

The authors have presented their review about Pain & Stress detection using Wearable Sensors and devices

The authors have put forward a fine effort in presenting their review. The article highlights pain detection and management, and applications of wearables for the same.

It is generally difficult to find novelty and innovations in a review article, and this article is not an exception. The review methodology is unclear and the objectives, goals, and purpose of this article are not explicitly understandable as the title and abstract seem to contradict. For example, the title suggests that this article is about a review of wearable sensors for pain / stress detections, whereas, the article is mostly about pain management with very little emphasis on the wearable sensors or at-least the non-invasive methods.

Below are my comments to the authors; and in the view of maintaining the scope and quality of the Journal, there are a few major concerns that must be addressed before this article can be accepted for publication.

Below are the comments and suggestions to improve the article:

  1. The presence of numerous other articles (original research / methodological reviews / systematic reviews / other reviews) limits this article’s relevance and novelty. More precisely, this article can be more comprehensive and should be expanded further to include relevant information and cover all the bases that touch upon the wearables. For instance, the wearables part consists of around 30% percent of the whole article. This should be expanded substantially since this is a review article. The scope and extent of the comprehensiveness and coverage could be improved to make this article stand out from the other similar reviews. This will provide the readers with a more in-depth understanding and clarity.
  2. As indicated in the multiple references within the article and numerous other articles published already, there are numerous other approaches to review the similar scope of work. Please include a section to emphasize how the approach of this review is different from others.
  3. Please make sure all the figures have references/citations. (throughout the article)
  4. Please make sure references/sources are adequate and correctly used throughout the article. (throughout the article)
  5. Please review usage of abbreviations/acronyms, and provide an explanation where required. (throughout the article)
  6. The article is missing a section about review methodology. A review methodology section is required to show how the literature review was conducted with an emphasis on eligibility criteria of the articles reviewed vs. the ones included in this review, search strategy, screening procedure, quality assessment, etc.
  7. There are several grammatical errors, dangling modifiers, sentence structure errors, etc throughout the article. The whole article should be thoroughly revised and proofread to rectify these errors. (throughout the article)
  8. Please rewrite sections 7 and 8 (discussion and conclusion) to reflect the above-mentioned corrections.

Authors responds:

  1. The presence of numerous other articles (original research / methodological reviews / systematic reviews / other reviews) limits this article’s relevance and novelty. More precisely, this article can be more comprehensive and should be expanded further to include relevant information and cover all the bases that touch upon the wearables. For instance, the wearables part consists of around 30% percent of the whole article. This should be expanded substantially since this is a review article. The scope and extent of the comprehensiveness and coverage could be improved to make this article stand out from the other similar reviews. This will provide the readers with a more in-depth understanding and clarity.

Ans:

Thank you for your comment. We have added a new section to brought into more discussion about wearable sensors in lines 526-568 and added a new figure (i.e., Figure 1 at line 525) to improve the comprehensiveness and coverage of this article.

  1. As indicated in the multiple references within the article and numerous other articles published already, there are numerous other approaches to review the similar scope of work. Please include a section to emphasize how the approach of this review is different from others.

Ans:

Thank you for your comment. We have added new segment in lines 48-60 to emphasize methodology approach and the purpose of this review article.

  1. Please make sure all the figures have references/citations. (throughout the article)

Ans:

Thank you for your comment. We have added references and checked again to make sure every references and citations are properly included throughout the article.

  1. Please make sure references/sources are adequate and correctly used throughout the article. (throughout the article)

Ans:

Thank you for your comment. We have checked again to make sure every references and sources are correctly used in this article.

  1. Please review usage of abbreviations/acronyms, and provide an explanation where required. (throughout the article)

Ans:

Thank you for your comment. We have checked again to make sure every abbreviations or acronyms are explained and provided in the entire article.

  1. The article is missing a section about review methodology. A review methodology section is required to show how the literature review was conducted with an emphasis on eligibility criteria of the articles reviewed vs. the ones included in this review, search strategy, screening procedure, quality assessment, etc.

Ans:

Thank you for your comment. We have added a new section in lines 61-78 to highlight the different review methodology approach of this review article to others.

  1. There are several grammatical errors, dangling modifiers, sentence structure errors, etc throughout the article. The whole article should be thoroughly revised and proofread to rectify these errors. (throughout the article)

Ans:

Thank you for your comment. The entire article has been checked again by all of the authors to rectify the linguistic errors. Particularly, one of our authors (i.e., Dr. Maysam F. Abbod) who is a senior reader in Brunel University London and has been living in UK for over 25 years. He has checked this revised manuscript not only the structure and contents but also line by line for English grammar and spelling.

  1. Please rewrite sections 7 and 8 (discussion and conclusion) to reflect the above-mentioned corrections.

Ans:

Thank you for your comment. Discussion and conclusion section are edited to accommodate the modification to the above suggestions in lines 581-585 and 616-624.

Round 2

Reviewer 1 Report

After the revision, we think this review can be published.

Author Response

Comments of reviewer I:

After the revision, we think this review can be published.

Authors responds:

Thank you for your recommendation.

Reviewer 3 Report

The authors have rectified the errors and/or have addressed the suggestions and corrections cited in the previous review.

I recommend this article to be accepted for publication pending any editorial and/or language revisions. 

Author Response

Comments of reviewer III:

The authors have rectified the errors and/or have addressed the suggestions and corrections cited in the previous review.

I recommend this article to be accepted for publication pending any editorial and/or language revisions. 

Authors responds:

Thank you for your recommendation. The manuscript has been checked again by all of the authors to eliminate any concerns of linguistic errors.